# Characterising a Weight Loss Intervention in Obese Asthmatic Children

**DOI:** 10.3390/nu12020507

**Published:** 2020-02-17

**Authors:** Shaun Eslick, Megan E. Jensen, Clare E. Collins, Peter G. Gibson, Jodi Hilton, Lisa G. Wood

**Affiliations:** 1Priority Research Centre for Healthy Lungs, Hunter Medical Research Institute, The University of Newcastle, New Lambton Heights, NSW 2305, Australia; shaun.eslick@uon.edu.au (S.E.); peter.gibson@newcastle.edu.au (P.G.G.); 2Priority Research Centre Grow Up Well, Hunter Medical Research Institute, School of Medicine and Public Health, The University of Newcastle, New Lambton Heights, NSW 2305, Australia; megan.jensen@newcastle.edu.au; 3Priority Research Centre in Physical Activity and Nutrition, Faculty of Health and Medicine, The University of Newcastle, Callaghan, NSW 2308, Australia; clare.collins@newcastle.edu.au; 4Pediatric Respiratory and Sleep Medicine, John Hunter Children’s Hospital, New Lambton Heights, NSW 2305, Australia; jodi.hilton@health.nsw.gov.au

**Keywords:** weight loss, asthma, children, diet, nutritional biomarkers

## Abstract

The prevalence of obesity in asthmatic children is high and is associated with worse clinical outcomes. We have previously reported that weight loss leads to improvements in lung function and asthma control in obese asthmatic children. The objectives of this secondary analysis were to examine: (1) changes in diet quality and (2) associations between the baseline subject characteristics and the degree of weight loss following the intervention. Twenty-eight obese asthmatic children, aged 8–17 years, completed a 10-week diet-induced weight loss intervention. Dietary intake, nutritional biomarkers, anthropometry, lung function, asthma control, and clinical outcomes were analysed before and after the intervention. Following the intervention, the body mass index (BMI) z-score decreased (Δ = 0.18 ± 0.04; *p* < 0.001), %energy from protein increased (Δ = 4.3 ± 0.9%; *p* = 0.002), and sugar intake decreased (Δ = 23.2 ± 9.3 g; *p*= 0.025). Baseline lung function and physical activity level were inversely associated with Δ% fat mass. The ΔBMI z-score was negatively associated with physical activity duration at baseline. Dietary intervention is effective in achieving acute weight loss in obese asthmatic children, with significant improvements in diet quality and body composition. Lower lung function and physical engagement at baseline were associated with lesser weight loss, highlighting that subjects with these attributes may require greater support to achieve weight loss goals.

## 1. Introduction 

Over the past few decades, the prevalence of both obesity and asthma has increased significantly worldwide in children [1]. Asthma is the most common chronic childhood disease and was estimated in 2007–2008 to affect 10.4% of children aged 0–15 years in Australia [2,3]. Obesity is a worldwide epidemic, with a study of 188 countries reporting 23.8% of boys and 22.6% of girls aged 2–19 years are overweight or obese in developed countries [4]. Australian data from 2014–2015 found that 27% of children aged 5–14 years were overweight and an additional 7% were obese [5]. 

Childhood obesity has been identified as a strong predictor of obesity in adulthood, with 6.4% of males and 12.6% of females carrying obesity from childhood to adulthood [6]. Of concern are a myriad of obesity-related respiratory problems and, in adults, these include mechanical lung compression, resistance to steroid treatment, increased systemic inflammation, and altered airway inflammation [7,8,9,10]. In children, excess weight is associated with poorer asthma control, increased risk of exacerbations, reduced effectiveness of steroids, and decreased static lung function, implicating the need for obesity management in paediatrics [10,11,12]. Therefore, addressing obesity in children with asthma is of high importance, not only to improve short-term health, but also long-term health.

To date, few weight loss studies utilising a dietary intervention alone have been conducted in obese children with asthma. The present study is a secondary analysis of a trial by Jensen et al. [13], a randomised controlled trial of a short-term dietary intervention in obese asthmatic children, which achieved improvements in lung function and asthma control as a result of an average weight loss of 3.4 kg. Subsequently, there have been three weight loss studies undertaken in asthmatic children that have demonstrated improvements in asthma control and severity, quality of life, static lung function, and fewer acute asthma exacerbations and nocturnal symptoms following weight loss of 2.6–13% [13,14,15,16]. Considering the reported benefits of weight loss in childhood asthma, studies examining the various strategies used are warranted.

Therefore, the aims of the current study were: (1) To examine changes in diet quality in obese asthmatic children during a weight loss intervention, and (2) To examine the association between the baseline subject characteristics and degree of weight loss following the intervention.

## 2. Materials and Methods

### 2.1. Study Design

This is a secondary analysis of a group of obese children with physician-diagnosed asthma who participated in a 10-week dietary intervention trial, which has been previously described [13]. Briefly, obese children (body mass index (BMI) z-score ≥ 1.64 standard deviation score (SDS)), aged between 8–17 years, with stable asthma (defined as no exacerbation, respiratory tract infection, oral corticosteroid use, or change in asthma medications in the past 4 weeks) were recruited from the John Hunter Children’s Hospital (JHCH) outpatient clinics, local medical centres, and the general community in Newcastle, Australia. Exclusion criteria for this study included: unexplained weight change during the past 3 months, inflammatory or endocrine disorders, and respiratory disorders other than asthma. Participants were randomised to either the dietary intervention group (DIG) or the wait-list control group (WLC), who received the same intervention as the DIG group after an initial 10 week waiting period. As the degree of weight loss was similar in the DIG and WLC groups, these were combined for this secondary analysis. Participant approval and guardian consent were acquired prior to enrolment. The study was registered with the Australian New Zealand Clinical Trials Registry (ACTRN12610000955011) and was approved by the Hunter New England and University of Newcastle Human Research Ethics Committees (09/05/20/5.08).

### 2.2. Intervention

The 10-week dietary intervention pursued a 2000-kilojoule/day (KJ/day) energy reduction from individually calculated age- and gender-appropriate energy requirements (Schofield equation to estimate basal metabolic rate using activity factor of 1.55) [17]. Participants attended face-to-face counselling sessions with an Accredited Practising Dietitian in weeks 0, 1, 2, 4, 6, 8, and 10 with telephone contact in alternate weeks. Counselling sessions involved theoretical and practical education on food selection as well as appropriate serving sizes to optimize macronutrient and micronutrient intakes within an energy-restricted diet, identification and resolution of barriers to dietary change, and goal-setting. Materials included individually adapted meal plans and a commercial calorie counter. Meal plans routinely encouraged participants to increase intake of wholegrain breads and cereals, fruit and vegetables, and low-fat dairy products and lean meats. Additionally, intake of foods high in excess energy, fat, sugar, and salt such as chips, pizza, sausage rolls, cakes, soft drinks, and fried foods such as chicken nuggets and hot chips, were discouraged. Participants and their guardians were instructed to self-monitor energy intake using a food diary throughout the study period. 

### 2.3. Clinical Assessment

At baseline and post-intervention, participants attended John Hunter Children’s Hospital after an overnight fast (≥12 h) and withholding antihistamines and asthma medications (≥24 h). Baseline clinical asthma pattern (Global Initiative for Asthma (GINA) guidelines) and atopy (skin prick test) were assessed as described previously [18]. The following data were assessed at baseline and post-intervention: asthma control (Juniper Asthma Control Questionnaire (ACQ)), quality of life (Paediatric Asthma Quality of Life Questionnaire (standardized) (PAQLQ(s))), dynamic and static lung function via spirometry (Windows KoKo PFT System Version 4.9 2005, PDS Inc., Louisville, KY, USA), and plethysmography (MedGraphics Elite Series Plethysmograph, St. Paul, MN, USA; Breeze Suite 6.4.1.14 Version 510 2008, MedGraphics Corp., St. Paul, MN, USA) [19,20]. Forced expiratory volume in one second (FEV1), forced vital capacity (FVC), and expiratory reserve volume (ERV) values were expressed as a percentage of the predicted values [21,22].

### 2.4. Anthropometry

Height and weight were measured at baseline and post-intervention using 150 kg max scales (EB8271 NuWeigh, Newcastle Weighing Services, Newcastle, NSW, Australia) and a 2 m wall-suspended measuring tape with wall stop (Surgical and Medical Supplies Pty Ltd., Rose Park, SA, Australia). Waist circumference measurement was collected using a tape measure (Lufkin Executive Thin line 2 m W606 PM tape measure). BMI was calculated (weight (kg)/height (m^2^)) and converted to BMI z-scores [23]. Body composition, including total body fat and lean mass, was measured as a percentage (%) of total body weight by dual energy X-ray absorptiometry (DEXA) (GE Lunar Prodigy, Medtel, Madison, WI, USA; GE Healthcare encore 2007 software Version 11.40.004, Madison, WI, USA) at baseline and post-intervention. 

### 2.5. Diet Quality

#### 2.5.1. Dietary Intake

Dietary intake was estimated pre- and post- intervention using a 4-day food record completed by participants using household measures. Records were analysed using the AUSNUT 2010 database available on nutrient analysis software Foodworks (Foodworks version 7.0.3016, Xyris Software, Brisbane, QLD, Australia) to quantify macronutrient and micronutrient intake. Nutrient intake was compared to the age and gender specific Nutrient Reference Values (NRVs) for Australians [24]. 

#### 2.5.2. Nutritional Biomarkers

##### Plasma Carotenoid and Tocopherol Analysis

High performance liquid chromatography (HPLC) methodology was used to determine β-carotene, lycopene, α-carotene, β-cryptoxanthin, lutein/zeaxanthin, α-tocopherol, and γ-tocopherol concentrations in plasma, as described previously [25,26]. 

##### Plasma and Red Blood Cell Fatty Acid Analysis

Gas chromatography (GC) was used to determine red blood cell (RBC) fatty acid (FA) proportions and plasma FA concentrations of saturated fatty acids (SFA), polyunsaturated fatty acids (PUFA), monounsaturated fatty acids (MUFA), and omega 3 and 6. RBC membrane and plasma FAs were methylated, and the total FAs were determined using the validated method established by Lepage and Roy as described previously [27,28]. 

### 2.6. Physical Activity

Baseline physical activity measurements, including the amount of exercise (duration) undertaken and the intensity of physical activity as measured by metabolic equivalent (METs) for subjects, was obtained via the Adolescent Physical Activity and Recall Questionnaire (APARQ). 

### 2.7. Statistical Analysis

Data are presented as mean (standard deviation, SD), median (interquartile range, IQR), or proportion (n, (%)). Continuous data were assessed using a paired mean-comparison *t*-test or Wilcoxon sign-rank test for within-group comparisons. Correlation analysis was conducted using Pearson’s correlation coefficient or Spearman’s Rank correlation coefficient to identify variables that correlated with weight loss, indicated by change in (Δ) BMI z-score and %fat mass. Results were considered statistically significant at *p* < 0.05. Statistical analysis was performed using Statistical Software for the Social Sciences Version 24.0 (SPSS Release 24.0; IBM Corp., Armonk, NY, USA). No adjustments were made for multiple testing. 

## 3. Results

### 3.1. Participant Characteristics

Baseline characteristics for the 28 participants are presented in Table 1. Participants were predominantly mild asthmatics, with normal lung function between 80–120% predicted values. Baseline characteristics revealed that participants were predominantly male and atopic. 

### 3.2. Changes Following The Intervention 

#### 3.2.1. Lung Function and Medication Use

No significant change in lung function or medication use were observed following the intervention [13].

#### 3.2.2. Anthropometric and Body Composition

Complete pre- and post-intervention anthropometric and body composition data were available for 27 participants. Following the intervention, a significant reduction in various anthropometric and body composition measurements (*p*-value ≤ 0.001) were observed. Furthermore, a significant increase in lean mass (1.9%) was observed following the intervention (Table 2).

#### 3.2.3. Diet Quality

##### Dietary Change

Complete pre- and post-intervention diet quality data were available for 16 participants. A significant increase in mean % energy derived from protein (16.2 ± 3.1 versus 20.5 ± 3.7, *p* < 0.001) and a significant decrease in absolute sugar intake (g) (115.3 ± 34.6 versus 92.2 ± 29.4, *p* = 0.025) were detected post intervention (Figure 1). A trend towards decreased intake of energy and % energy from fat was observed (Table 3). At baseline, the mean intake of fibre, Vitamin A, potassium, and calcium were below age and gender specific recommendations. Post-intervention, the mean intake of fibre, Vitamin A, potassium, and calcium intake remained inadequate. The intake of saturated fat as % total fat intake was relatively high, approximately 38%, and remained unchanged post-intervention. 

##### Nutritional Biomarkers

No significant change in plasma carotenoid and tocopherol concentrations was detected following the intervention (Table 4). No significant differences in total red blood cell membrane fatty acids or individual fatty acids was seen in participants following the intervention (Table 5). 

### 3.3. Correlations 

The Δ% fat mass was negatively associated with baseline % predicted FEV1 (*r* = −0.429, *p* = 0.026) (Figure 2 a) and baseline METs (*r* = −0.454, *p* = 0.023) (Figure 2b). Baseline duration of physical activity (mins/week) was negatively associated with ΔBMI z-score (*r* = −0.445, *p* = 0.023) (Figure 3).

## 4. Discussion

In this secondary analysis of a cohort of obese asthmatic children who participated in a 10-week diet-induced weight loss intervention, we evaluated diet quality, which was assessed by dietary intake and key nutritional biomarkers, and examined the association between baseline subject characteristics with degree of weight loss following the intervention [13]. Following the intervention, improvements in diet quality were observed, notably a significant increase in protein intake and a significant decrease in sugar consumption. No significant changes in micronutrient intake, plasma carotenoids, or fatty acids were observed. Interestingly, children who had better baseline lung function (%predicted FEV1), or who undertook higher intensity physical activity at baseline, had a greater loss of % body fat. Additionally, children who engaged in a longer duration of physical activity at baseline had a greater decrease in BMI z-score. 

The dietary intervention was successful in inducing acute weight loss in this group of obese asthmatic children, with a mean 3.4% weight loss and a 0.18 SDS reduction in the BMI z-score over 10 weeks. This is comparable to previous work reporting a 2.6% weight loss and a 0.2 SDS reduction in the BMI z-score in overweight/obese asthmatic children who undertook a dietary intervention over 6 weeks [15]. Of recent weight loss studies in non-asthmatic children that included a dietary component, BMI z-score reductions of 0.1–0.4 have been reported in interventions conducted over longer periods of time, lasting 4–9 months, whilst other studies found no significant change in BMI z-score [29,30]. Our results also show a significant decrease in the mean waist circumference of 3.7 cm compared to a 6 cm reduction in waist circumference in non-asthmatic overweight children involved in a multi-faceted lifestyle intervention conducted over a much longer 6-month period by Reinehr et al. [31]. Lastly, significant reductions in fat mass were also observed in our study, with a mean fat loss of 2.2% and significant reductions in segmental fat. A multicomponent lifestyle intervention study by da Silva et al. in asthmatic adolescents also reported significant fat loss with a reduction of 6%, albeit over a longer period of 6 months [14]. The lean mass of participants in our study significantly increased by 1.9% compared to a non-significant improvement in lean mass between control and intervention groups following the 6-month intervention carried out by Reinehr et al. [31]. This is a notable finding in the absence of a structured exercise component. Improvement seen in lean mass may have been attributable to the main dietary findings of increased protein and decreased sugar intake following the intervention. Therefore, the study outcomes suggest the efficacy of this short-term dietary intervention in reducing weight loss, BMI z-score, waist circumference, and fat mass in obese asthmatic children. 

Participants had a significant increase of 4.3% in energy derived from protein and a 20% reduction in absolute sugar intake. Additionally, a trend of decreased % energy from total fat (−3.1%) was observed; however, this did not reach statistical significance. The observed dietary changes as a result of the intervention are likely attributable to the nutrition strategies employed in the intervention, such as meal plans and portion size education, which discouraged large intakes of foods high in excess energy, fat, sugar, and salt. Similarly, studies by Davis et al. [32,33] in non-asthmatic overweight adolescent females that focused on inducing weight loss through energy restriction achieved significant decreases in sugar intake and trended towards increased % energy intake from protein, mainly achieved by reducing the intake of processed foods. Therefore, our data demonstrate that an intervention spanning 10 weeks can induce important dietary changes; however, it is unknown if these changes can be maintained long term, and longer-term studies in this population are warranted. Additionally, the use of electronic data collection, such as a mobile phone application to record dietary intake, may reduce the burden of completing a 4-day record and increase compliance, particularly in this age group [34,35]. 

Dietary records revealed that baseline consumption of various micronutrients: vitamin A, fibre, calcium, and potassium, was inadequate when compared to the NRVs, and remained inadequate following the intervention. Interestingly, when compared to the 2011–2012 Australian health survey, inadequate intakes of calcium and Vitamin A were common in the average Australian child [36]. Approximately 70% of males and 87% of females aged 12–18 years were reported to have inadequate intakes of calcium [36]. Inadequate intake of vitamin A was seen in 33% of males and 27% of females aged 14–18 years [36]. Adequate intakes of these nutrients are particularly important in children for a range of essential body functions such as the use of Vitamin A to promote normal vision and a strong immunity to infectious disease, fibre to promote good bowel health, and calcium to support adequate growth and development. Therefore, it is concerning that intakes of these nutrients within this cohort are inadequate [24]. The study intervention did not specifically target these micronutrients; the focus was on dietary energy reduction and improving overall diet quality. Furthermore, the relatively short time frame of the intervention limited the amount of dietary change achievable. Studies inducing sustained increased intakes of fruit and vegetables in children have been conducted over 6–12 months, indicating that improving micronutrient intakes is possible in studies of longer duration [37,38]. Notably, intake of saturated fatty acids remained relatively high (38% of total fat intake) pre- and post-intervention. In dietary interventions of longer duration, this may be an important area to target as high saturated fat intake is associated with increased risk of cardiovascular disease [39,40]. Interestingly, vitamin E intake reduced following the intervention. A large number of polyunsaturated oils and processed foods are fortified with or contain vitamin E; therefore, a decreased intake of these foods as a result of dietary strategies may explain a decrease in the dietary intake of vitamin E [41]. 

Significant changes in plasma carotenoids, tocopherols, and fatty acid biomarkers were not detected following the 10-week weight loss intervention, despite improvements in dietary intake. To our knowledge, this is the first study in children with asthma that was conducted to examine the change in nutritional biomarkers following a weight loss intervention. Findings from nutritional biomarker analysis are consistent with the findings from 4-day food records, indicating that fruit, vegetable, and fat intake of participants did not significantly change as a result of the dietary intervention in this small sample. Future studies including longer dietary interventions, to promote micronutrient improvements as well as energy restrictions, are required in obese asthmatic children. Adequate nutrient intake in childhood, including fibre and vitamins A and E, is not only essential for optimal growth and development, but could also potentially provide beneficial anti-inflammatory and antioxidant effects in the asthmatic population [42,43,44]. 

An investigation of the baseline characteristics that were associated with the degree of weight loss in our study found that children who presented with better lung function at baseline achieved greater fat loss. This opposes results from a dietary intervention trial in obese asthmatic adults, whereby having poorer lung function was a positive predictor of weight loss [45]. Our results, which suggest that poorer asthma status is a barrier to lifestyle change in children, may be due to factors such as exercise avoidance and increased corticosteroid use in children with more severe disease [46,47]. Indeed, we found that more intense physical activity at baseline (METs) was correlated with greater % fat loss, and longer participation in physical activity at baseline was correlated with a greater reduction in the BMI z-score. Our data suggest that those with better lung function or physical engagement would be a good target for future interventions as they are more likely to achieve weight loss goals. In contrast, these findings suggest that children who do not have these attributes experience a barrier to weight loss and may benefit from a different type of intervention. Such an intervention may require extra assistance to achieve weight loss goals through asthma education on approaches to exercising safely. 

There are a few limitations that should be acknowledged. Firstly, the small sample size potentially limited our ability to detect significant changes in some outcomes; nonetheless, we had adequate power to detect key changes in weight status and diet quality, which provided novel and important data to stimulate research in this area of childhood obesity and asthma. Secondly, we do not have follow up data on participants; therefore, long-term diet quality, weight loss, or maintenance could not be examined. Lastly, due to a low response rate, analysis of 4-day food records was limited and reporting bias may have affected results. However, the assessment of diet quality was strengthened by the use of gold standard objective measurement tools such as high performance liquid chromatography and gas chromatography to analyse nutritional biomarkers in the majority of subjects. 

Analysis of diet quality revealed that a short term, diet-induced weight loss intervention was successful in reducing overall energy intake and intake of sugary foods; however, dietary data and nutritional biomarker analysis indicated no change in fruit and vegetable or fatty acid intake, perhaps due to a delay in the adoption of these positive practices. The study indicated that partaking in more regular or intense exercise at baseline was correlated with greater weight loss success, whilst those children with poorer lung function at baseline achieved less weight loss, so may require greater support to achieve their weight loss goals. The findings from this study support the need for future larger trials to further investigate the efficacy of weight loss interventions in obese asthmatic children. Future trials should include a longer intervention period to allow the implementation and adoption of more comprehensive dietary strategies targeting a reduction in dietary energy and improvements in diet quality, including an increase in fruit and vegetable intake and a reduction in saturated fatty acid intake. Importantly, this study has highlighted subgroups of obese asthmatic children, specifically those with low lung function or low physical activity levels, that may require additional support in order to succeed in weight loss interventions. 

## Figures and Tables

**Figure 1 nutrients-12-00507-f001:**
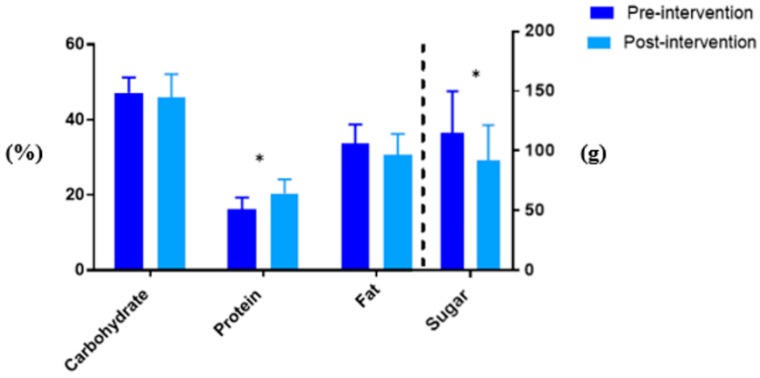
The % Energy intake from carbohydrate, protein, and fat as well as absolute sugar intake (g) pre- and post-intervention. * *p* < 0.05 for pre- versus post-intervention values.

**Figure 2 nutrients-12-00507-f002:**
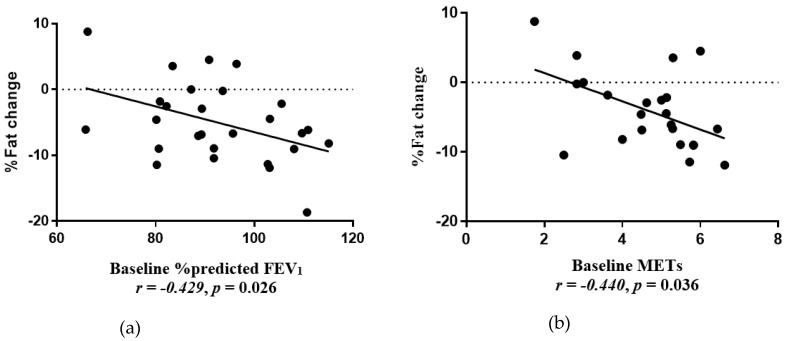
(**a**) Correlation between % fat mass change and baseline %predicted forced expiratory volume in 1 s (FEV_1_) (*r* = −0.429, *p* = 0.026). (**b**) Correlation between % fat mass change and baseline Metabolic Equivalents (METs) (*r* = −0.440, *p* = 0.036).

**Figure 3 nutrients-12-00507-f003:**
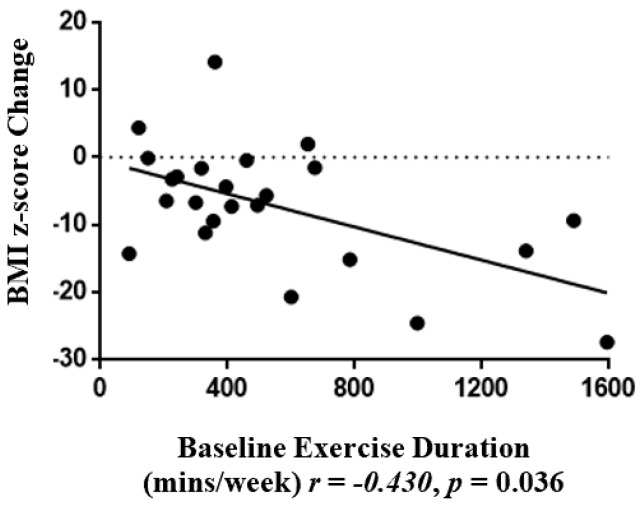
Correlations between BMI z-score change and baseline physical activity duration (mins/week) (*r* = −0.430, *p* = 0.036).

**Table 1 nutrients-12-00507-t001:** Subject characteristics at baseline.

Characteristics	Baseline
Subjects; n	28
Gender (%females)	11 (39.3)
Age (years)	12.1 ± 2.3
Height (cm)	156.6 ± 12.2
Weight (kg)	73.3 ± 3.9
BMI z-score	2.1 ± 0.3
Waist Circumference (cm)	98.1 ± 10.6
Total body fat mass (%)	45.1 ± 6.8
Total lean mass (%)	53.3 ± 6.4
ACQ score	1.0 (0.4, 1.4)
PAQLQ score	6.1 (5.2, 6.5)
FEV1 %predicted	92.8 ± 2.4
FVC %predicted	100.5 (93.7, 108.4)
FEV1/FVC %	94.2 (88.1, 96.5)
ERV %predicted	94.6 (66.2, 147.9)
Atopic; n (%)	19 (67.9)
Metabolic Equivalent (METs)	5.2 (3.9, 5.8)
Activity Duration (mins)	436.3 (285.0, 702.5)
Short-acting Beta-antagonist; n (%)	24 (84.7)
Inhaled Corticosteroid; n (%)	10 (35.7)

Data are presented as mean ± SD or median (interquartile range, IQR) unless stated. BMI z-score: Body Mass Index z-score; ACQ: Asthma Control Questionnaire; PAQLQ: Paediatric Asthma Quality of Life Questionnaire; FEV1: Forced Expiratory Volume in 1 s; FVC: Forced Vital Capacity; ERV1: Expiratory Reserve Volume; METs: Metabolic Equivalent of Task; SABA: Short acting β-agonist; ICS: Inhaled Corticosteroids.

**Table 2 nutrients-12-00507-t002:** Change in anthropometric variables in obese asthmatic children following a 10-week dietary intervention.

Anthropometry and Body Composition	Pre-Intervention	Post-Intervention	*p*-Value
Weight (kg)	73.3 ± 20.71	70.8 ± 19.35	0.001
BMI z-score	2.13 ± 0.30	1.95 ± 0.31	<0.001
Waist circumference (cm)	98.1 ± 9.72	94.4 ± 9.05	<0.001
Total body fat mass (%)	45.1 ± 6.74	42.9 ± 7.38	<0.001
Lean mass (%)	53.1 ± 6.3	55.0 ± 7.2	<0.001

Data are presented as mean ± SD or median (QR) unless stated.

**Table 3 nutrients-12-00507-t003:** Change in dietary intakes in obese asthmatic children following a 10-week dietary intervention.

Energy and Macronutrients	Pre-Intervention	RDI/AI (%)	Post-Intervention	RDI/AI (%)	*p*-Value
Energy (kJ)	8678.6 ± 2089.9	-	7548.3 ± 866.6	-	0.123
%Protein	16.2 ± 3.1	-	20.5 ± 3.7	-	<0.001
%Fat	33.8 ± 5.0	-	30.7 ± 5.6	-	0.147
%SFA	38.4 ± 2.8	-	38.7 ± 3.9	-	0.794
%MUFA	12.7 (11.0, 18.0)	-	16.5 (12.7, 18.9)	-	0.109
%PUFA	48.6 (44.8, 51.4)	-	45.1 (40.9, 49.0)	-	0.121
%Carbohydrate	47.2 ± 4.1	-	45.9 ± 6.3	-	0.493
Sugar (g)	115.3 ± 34.6	-	92.2 ± 29.4	-	0.025
Fibre (g)	19.8 (16.5, 23.2)	89.5 (76.3, 113.4)	21.9 (15.7, 28.2)	99.4 (73.1, 122.2)	0.438
β-Carotene (µg)	2066.1 (836.2, 4436.1)	53.5 (23.2, 102.3)	2116.1 (1442.9, 3210.3)	52.9 (36.9, 94.5)	0.717
Thiamine (mg)	2.2 (1.7, 3.4)	229.6 (176.4, 318.7)	2.2 (1.6, 3.0)	214.9 (174.3, 338.9)	0.717
Riboflavin (mg)	2.7 (1.7, 3.2)	275.2 (190.1, 323.5)	2.5 (1.8, 3.0)	279.8 (194.1, 334.0)	0.959
Niacin (mg)	43.3 ± 12.4	361.6 ± 26.2	45.8 ± 16.0	395.1 ± 42.7	0.649
Vitamin C (mg)	95.9 (70.2, 117.8)	239.6 (175.4, 294.5)	61.2 (29.9, 91.3)	153.1 (74.6, 228.1)	0.179
Vitamin E (mg)	8.8 ± 5.3	102.1 ± 13.0	6.9 ± 2.2	81.8 ± 6.8	0.197
Potassium (mg)	2409.0 ± 509.1	78.0 ± 4.2	2570.2 ± 706.6	84.7 ± 7.2	0.497
Calcium (mg)	754.2 ± 245.3	66.6 ± 6.4	802.2 ± 385.7	72.1 ± 10.2	0.477
Iron (mg)	11.2 ± 3.6	127.6 ± 11.1	11.8 ± 5.2	134.9 ± 15.6	0.661
Zinc (mg)	9.3 (8.0, 13.0)	155.4 (127.2, 216.5)	12.2 (8.3, 14.5)	184.6 (116.2, 236.7)	0.756

Data are presented as mean ± SD or median (IQR) unless stated. kJ: kilojoules; RDI: Recommended Daily Intake; AI: Adequate Intake; - not applicable.

**Table 4 nutrients-12-00507-t004:** Change in plasma carotenoids and tocopherols in obese asthmatic children following a 10-week dietary intervention.

Nutritional Biomarker	Pre-Intervention	Post-Intervention	*p*-Value
**Carotenoids (mg/mL)**			
Lutein	171.0 (152.5, 250.8)	207.5 (138.5, 268.8)	0.349
β-Cryptoxanthin	220.5 (78.0, 543.0)	254.5 (146.8, 346.8)	0.896
Lycopene	64.5 (42.3, 168.5)	53.0 (36.3, 97.5)	0.653
α-carotene	21.0 (0.00, 35.3)	21.0 (10.8, 31.8)	0.463
β-carotene	311.0 (114.0, 568.8)	304.5 (114.0, 711.8)	0.352
Total Carotenoids	995.5 (528.5, 1504.3)	946.0 (494.3, 1494.3)	0.472
**Tocopherols (mg/mL)**			
γ-tocopherol	1.56 (0.93, 2.26)	1.95 (1.18, 2.41)	0.396
α-tocopherol	19.17 ± 2.41	23.56 ± 2.82	0.157
Total Tocopherols	20.88 (9.69, 29.46)	26.22 (18.23, 29.28)	0.157

Data are presented as mean ± SD or median [IQR] unless stated.

**Table 5 nutrients-12-00507-t005:** Change in red blood cell membrane fatty acids in obese asthmatic children following a 10-week dietary intervention.

Nutritional Biomarker	Pre-Intervention	Post-Intervention	*p*-Value
SFA %	44.0 (43.8, 45.3)	43.9 (43.8, 45.2)	0.326
PUFA %	38.0 ± 1.6	37.9 ± 1.1	0.366
MUFA %	17.6 ± 0.8	17.9 ± 0.7	0.205
Omega 3 %	5.4 (4.0, 7.2)	6.7 (5.0, 7.7)	0.281
Omega 6 %	43.9 (36.8, 49.0)	42.0 (37.0, 47.6)	0.379

Data are presented as mean ± SD or median (IQR) unless stated.

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
