# Peer review of "Characterising a Weight Loss Intervention in Obese Asthmatic Children"

_nutrients, 2020, doi:10.3390/nu12020507_

Round 1
Reviewer 1 Report
This manuscript focusses an important issue. It is well-known that obesity is a disease modifier for asthma in children and adults. Besides, obese asthmatic patients presents worse disease control with increased asthma severity and poor response to the therapies. The present study is an extension of a first study and sought to evaluate the effect of 10-week intervention in dietary parameters and nutritional biomarkers.
I made the following suggestions and questions:
Minor revisions
Abstract Lines 21 and 24 - Please replace “BMI-z score” by “BMI z-score”
Introduction Lines 41 and 42 - What do authors want to say with “significant lung restriction”?
Material and methods Lines 67 and 68 - Lack exclusion criteria according to the registered protocol Line 77 - Replace “2000-kilojoule/day” by “2000 kilojoule/day (KJ/day)” Line 84 - “commercial counter” - is it an app? Lines 88 and 89 - “Participants and their guardians were encouraged to self-monitor energy intake using a food diary throughout the study period” - I believe in a weight loss intervention study where the diet quality was an aim, the filling of food diary should be mandatory Lines 92 and 93 and Lines 95 and 96 - The square brackets should appear first than round brackets - please standardize this throughout the text Line 107 - Put m2 in superscript (m2) Line 139 - Please replace “BMI-z score” by “BMI z-score”
Results Table 1 - Subjects; n - put like this Table 1 - Please replace “BMI-z score” by “BMI z-score” inside the table and in line 149 Line 155 - Please replace “Anthropometric and Body Composition” by “Anthropometry and Body Composition” Line 170 - Please replace “fats” by “fat” Table 3 - Please change “B-carotene” by “β-carotene” Table 3 - Please add “Energy and macronutrients” as the same way you put a section “Vitamins” and “Minerals” Table 3 - Add abbreviations 3 Correlations - Add a space between FEV1 (r = -0.429)
Discussion 0. Discussion - Put just “4. Discussion” Line 7 of Discussion - Replace “was observed” by “were observed” Lines 10, 14, 16, 17, 30 and 86 of Discussion - Please replace “BMI-z score” by “BMI z-score” Line 48 - I do not understand “Australian child.44” - maybe there is a mistake Lines 99 and 100 - I do not understand the sentence. It is quite confused. And what about the gold-standard? Please explain that sentence and provide a new one
References
References are not in the Nutrients style
Regarding all the p-values please uniformize that. For me, as title could stay p-value but when you say p<0.05, do not put value. Put lowercase and italic or capital letter and italic. Always the same throughout the paper “Highlighted in bold” - in the tables it does not appear in bold
It would be interesting to evaluate biochemical and metabolic parameters before and after intervention. Nutritional status biomarkers as albumin, pre-albumin as well.
Author Response
Thank you for the helpful comment and opportunity to revise our manuscript. Please see responses below
Response to Reviewer 1:
Abstract Lines 21 and 24 - Please replace “BMI-z score” by “BMI z-score”
Response: Amendments to the abstract have been made
Introduction Lines 41 and 42 - What do authors want to say with “significant lung restriction”?
Response: Significant lung restriction referred to ‘mechanical lung compression’ and has been amended in the text.
Material and methods Lines 67 and 68 - Lack exclusion criteria according to the registered protocol Line 77 - Replace “2000-kilojoule/day” by “2000 kilojoule/day (KJ/day)” Line 84 - “commercial counter” - is it an app? Lines 88 and 89 - “Participants and their guardians were encouraged to self-monitor energy intake using a food diary throughout the study period” - I believe in a weight loss intervention study where the diet quality was an aim, the filling of food diary should be mandatory Lines 92 and 93 and Lines 95 and 96 - The square brackets should appear first than round brackets - please standardize this throughout the text Line 107 - Put m2 in superscript (m2) Line 139 - Please replace “BMI-z score” by “BMI z-score”
Response: Amendments as suggested have been made throughout the text.
Results Table 1 - Subjects; n - put like this Table 1 - Please replace “BMI-z score” by “BMI z-score” inside the table and in line 149 Line 155 - Please replace “Anthropometric and Body Composition” by “Anthropometry and Body Composition” Line 170 - Please replace “fats” by “fat” Table 3 - Please change “B-carotene” by “β-carotene” Table 3 - Please add “Energy and macronutrients” as the same way you put a section “Vitamins” and “Minerals” Table 3 - Add abbreviations 3 Correlations - Add a space between FEV1 (r = -0.429)
Response: Amendments as suggested have been made throughout the text.
Discussion 0. Discussion - Put just “4. Discussion” Line 7 of Discussion - Replace “was observed” by “were observed” Lines 10, 14, 16, 17, 30 and 86 of Discussion - Please replace “BMI-z score” by “BMI z-score” Line 48 - I do not understand “Australian child.44” - maybe there is a mistake Lines 99 and 100 - I do not understand the sentence. It is quite confused. And what about the gold-standard? Please explain that sentence and provide a new one
Response: Amendments as suggested have been made. Line 48 has been reworded with the aim of explaining the comparison of nutrient intakes of study participants with the common Australian child. Line 99-100 has also been expanded to refer to specific gold standard techniques.
References are not in the nutrients style
Response: References list has been updated to nutrients/MDPI format using the download format for endnote function
Regarding all the p-values please uniformize that. For me, as title could stay p-value but when you say p<0.05, do not put value. Put lowercase and italic or capital letter and italic. Always the same throughout the paper “Highlighted in bold” - in the tables it does not appear in bold
Response: p-value has been amended to “p-value” throughout text. The references to bold have been removed.
It would be interesting to evaluate biochemical and metabolic parameters before and after intervention. Nutritional status biomarkers as albumin, pre-albumin as well.
Response: We agree that this would be an interesting addition, however, unfortunately this data was not collected.
Reviewer 2 Report
This manuscript focused on the characterising a weight loss intervention in obese asthmatic children. Overall, this manuscript is well-written and provides some interesting points. The manuscript is clearly written and well referenced. Each section is relevant and logically covers the topic. However, I have some suggestions to improve the manuscript: 1. We encourage author provided a summary table, listing all previous report about diet controled for asthma children. And, also summarized current clinical trials for this topic. 2. In this study, the children used some medicine, incluidng short-acting beta-antagonist and inhaled corticosteroid. During the diet intervention, any medicine change or shifted was record. 3. Authors has to report post-intervention pulmonary function test after diet controlled. 4. We encourage author provided a study design flow chart to make reader easily umderstand the design and result
Author Response
Thank you for the helpful comment and opportunity to revise our manuscript. Please see responses below
We encourage author provided a summary table, listing all previous report about diet controled for asthma children. And, also summarized current clinical trials for this topic.
Response: Thank you for your suggestion. We have described the other trials in this area in the Introduction and Discussion of the paper, however we believe that a summary table would not be a suitable addition to this original research article.
In this study, the children used some medicine, incluidng short-acting beta-antagonist and inhaled corticosteroid. During the diet intervention, any medicine change or shifted was record.
Response: Medicine usage was monitored and there were no changes to medication over the course of the intervention. A sentence has been added to the Results section to explain this:
‘3.2.1 Lung Function and Medication Use
No significant change in lung function or medication use were observed following the intervention.’
Authors has to report post-intervention pulmonary function test after diet controlled
Response: Lung function data was included in the primary paper, hence we have not included this data, to avoid repeating information that has already been published. A sentence has been added to the Results, explaining that there were no changes in lung function following the intervention:
‘3.2.1 Lung Function and Medication Use
No significant change in lung function or medication use were observed following the intervention.’
We encourage author provided a study design flow chart to make reader easily umderstand the design and result
Response: A study design flow chart has been included in the primary paper. To avoid repeating information that has been previously published, we have included a reference to this paper in the Methods:
‘This is a secondary analysis of a group of obese children with physician-diagnosed asthma who participated in a 10 week dietary intervention trial, which has been previously described [13].’